# Repositioning of Anti-Diabetic Drugs against Dementia: Insight from Molecular Perspectives to Clinical Trials

**DOI:** 10.3390/ijms241411450

**Published:** 2023-07-14

**Authors:** Keren Esther Kristina Mantik, Sujin Kim, Bonsang Gu, Sohee Moon, Hyo-Bum Kwak, Dong-Ho Park, Ju-Hee Kang

**Affiliations:** 1Department of Pharmacology, Research Center for Controlling Intercellular Communication, College of Medicine, Inha University, Incheon 22212, Republic of Korea; kerenmantik@inha.edu (K.E.K.M.); sujin2419@inha.ac.kr (S.K.); ooys0603@inha.edu (B.G.); moon219@inha.ac.kr (S.M.); 2Program in Biomedical Science and Engineering, Inha University, Incheon 22212, Republic of Korea; kwakhb@inha.ac.kr (H.-B.K.); dparkosu@inha.ac.kr (D.-H.P.); 3Department of Kinesiology, College of Arts and Sports, Inha University, Incheon 22212, Republic of Korea

**Keywords:** dementia, drug repositioning, anti-diabetics, insulin, insulin-like growth factor 1, clinical trial

## Abstract

Insulin resistance as a hallmark of type 2 DM (T2DM) plays a role in dementia by promoting pathological lesions or enhancing the vulnerability of the brain. Numerous studies related to insulin/insulin-like growth factor 1 (IGF-1) signaling are linked with various types of dementia. Brain insulin resistance in dementia is linked to disturbances in Aβ production and clearance, Tau hyperphosphorylation, microglial activation causing increased neuroinflammation, and the breakdown of tight junctions in the blood–brain barrier (BBB). These mechanisms have been studied primarily in Alzheimer’s disease (AD), but research on other forms of dementia like vascular dementia (VaD), Lewy body dementia (LBD), and frontotemporal dementia (FTD) has also explored overlapping mechanisms. Researchers are currently trying to repurpose anti-diabetic drugs to treat dementia, which are dominated by insulin sensitizers and insulin substrates. Although it seems promising and feasible, none of the trials have succeeded in ameliorating cognitive decline in late-onset dementia. We highlight the possibility of repositioning anti-diabetic drugs as a strategy for dementia therapy by reflecting on current and previous clinical trials. We also describe the molecular perspectives of various types of dementia through the insulin/IGF-1 signaling pathway.

## 1. Introduction

Dementia is a debilitating condition that affects millions of people worldwide. Currently, the available treatments target only specific pathological hallmarks, such as beta-amyloid (Aβ) in Alzheimer’s disease (AD), with the addition of symptomatic relief medications. However, dementia has multiple subtypes and diverse pathological mechanisms. As a result, there is a growing need to identify new therapeutic approaches that can effectively treat and prevent dementia. One promising strategy is repurposing existing drugs with potential in preclinical and clinical studies [1]. The brain was initially thought to be a non-insulin-sensitive organ. However, it is now acknowledged as an insulin-sensitive organ due to a growing body of evidence in neurodegenerative research. The emerging understanding of the brain’s insulin sensitivity has shed light on the potential link between insulin-related conditions and dementia [2]. Diabetes mellitus (DM), one of the most prevalent metabolic diseases, is associated directly or indirectly with increasing the risk for dementia. Based on the World Alzheimer Report 2021, more than 55 million people worldwide have dementia. This number of patients is expected to increase as the elderly population of developed countries, in particular, continuously increases [3]. Similarly, the number of people with diabetes is also experiencing a steady increase. According to the International Diabetes Federation (IDF), there are 536.6 million diabetics worldwide. This population is predicted to surpass 700 million by 2045 [4]. A population-attributable risk study using existing meta-analyses has shown that diabetes accounted for 2.9% of AD cases worldwide in 2010 [5]. Another study also agrees with this study by stating that a longer duration of diabetes is associated with an increased risk of dementia’s progression [6].

To date, aducanumab and lecanemab are the two approved disease-modifying therapies (DMT) for AD. Both are classified as human monoclonal antibodies that target the Aβ pathology in AD [7,8,9,10]. Multiple clinical trials aimed at DMT to target the primary pathogenic proteins implicated in AD pathogenesis, such as Aβ and Tau, have failed to halt AD progression. This occurrence can be attributed to the current limitations in the knowledge of how to effectively overcome the blood–brain barrier (BBB) when developing novel treatments for central nervous system (CNS) disorders [11]. Targeting specific pathologies in dementia treatment can be challenging, as neurodegenerative diseases often share interrelated pathologies. Therefore, multi-target drugs that can address multiple pathologies simultaneously may be preferable. Related to this, an ongoing study that investigates the efficacy of buntanetap, an orally bioavailable small molecule that suppresses the translation of mRNAs associated with neurotoxic aggregating proteins such as amyloid precursor protein, Tau, and α-synuclein, has received the FDA’s consent to proceed to the phase III clinical study [12]. The development of multi-target drugs for dementia treatment aligns with the concept of repositioning anti-diabetic drugs for dementia, as the insulin/IGF-1 signaling pathway has been implicated in the majority of dementia subtypes. From a molecular viewpoint, repositioning anti-diabetic drugs for DMT against multiple subtypes of dementia appears to be a reasonable approach. However, previous and ongoing clinical trials could not yet provide reliable outcomes.

In this paper, we highlight the possibility of repositioning anti-diabetic drugs as a strategy for dementia therapy by analyzing clinical trials as well as integrating the molecular perspectives from various types of dementia through the insulin/IGF-1 signaling pathway.

## 2. Insulin/Insulin-like Growth Factor 1 (IGF-1) Signaling in the Brain

### 2.1. The Source of Insulin and IGF-1 in the Brain

Most of the circulating insulin enters the brain primarily by insulin transport, which is made possible by insulin receptors (IR). Circulating insulin can be transported into the brain by binding to the IR found on the brain endothelial cells within the BBB, followed by internalization via endocytosis. The subsequent process of internalized insulin is not fully understood. It is unclear whether it is transcytosed or directly degraded. Nonetheless, the IR in brain endothelial cells appears critical for insulin transport, as demonstrated by studies of endothelial-specific knockout of the IR in mice. This knockout reduced IR phosphorylation in key brain regions, including the hypothalamus, hippocampus, and prefrontal cortex [13]. Studies have suggested that the brain may be capable of synthesizing small amounts of insulin. However, it is important to note that insulin production within the brain is minimal, and only low levels of insulin mRNA have been detected in specific brain regions, such as the anterior hypothalamus [14]. IGF-1 is capable of passing through the BBB in a similar manner to insulin. IGF-1 can cross the BBB by binding to IGF-1 receptors on the brain endothelial cells [15]. Contrary to insulin, IGF-1 mRNA has appeared to be highly expressed within the brain [16]. Researchers suggest that IGF-1 can be produced in the brain by neurons and glial cells, although the precise mechanisms are not entirely elucidated yet [17,18,19].

Restrictive components of the BBB should be considered when elucidating the mechanism of insulin or IGF-1 transport via BBB. These components include tight intercellular junctional complexes and efflux pumps. Intercellular junctional complexes refer to structures between adjacent brain endothelial cells that limit diffusion across the BBB. Efflux pumps, which are members of the ATP-binding cassette family, hydrolyze ATP to pump substances back into the plasma, thereby potentially limiting the amounts of substances that can enter the brain [13,20].

Differentiating between insulin/IGF-1 produced in the brain and those transferred via BBB presents a challenge due to their similarity in chemical identity. Anti-insulin antibodies recognize the identical epitopes of pancreatic and brain-derived insulin, making anti-insulin immunocytochemistry or radioimmunoassay ineffective [21]. Another way to identify insulin synthesis in the brain is to look for mRNA of insulin-coding genes such as Ins1 and Ins2 in mice, Ins2 in rats, or INS in humans [22,23]. The INS gene in humans is predominantly expressed in pancreatic beta cells, where it undergoes post-translational modifications to produce mature insulin. However, as mentioned earlier, small foci of insulin mRNA have been detected in certain brain regions, including the anterior hypothalamus. This result suggests that INS gene expression can occur in certain neurons within the brain, although the mechanisms underlying this local insulin synthesis are yet to be fully discovered [14]. As for IGF-1, one possible way to distinguish between IGF-1 produced in the brain and IGF-1 produced in other tissues is to examine the levels of IGF-1 in the cerebrospinal fluid (CSF) [24].

### 2.2. Neuroprotective Effect of Insulin and IGF-1

Insulin and IGF-1 play critical roles in regulating glucose metabolism in the brain. These hormones are important for promoting glucose uptake and utilization by neurons, which is critical for maintaining energy homeostasis and preventing neuronal damage and death. Additionally, insulin and IGF-1 can regulate the activity of enzymes involved in glucose metabolism, such as glycogen synthase kinase-3 (GSK-3). Dysregulation of GSK-3 has been implicated in the development of neurodegenerative diseases such as AD [25]. Therefore, the ability of insulin and IGF-1 to regulate GSK-3 activity may be an important mechanism for their neuroprotective effects. Previous studies have provided evidence for the critical role of insulin and IGF-1 in regulating glucose metabolism in the brain. A study demonstrated that impaired insulin signaling in the brain leads to reduced glucose uptake and metabolism through the deficiency of low-density lipoprotein receptor-related protein 1 (LRP1), a protein contributing to the removal of Aβ from the brain to circulation [26]. Another study showed that IGF-1 receptor-deficient mice showed reduced glucose uptake [27].

Insulin and IGF-1 have been shown to have anti-inflammatory effects on the brain, which can help protect against neurodegeneration. These growth factors can inhibit the activation of microglia and reduce the production of proinflammatory cytokines. Microglia, resident immune cells in the brain, play a key role in the inflammatory response in the brain. Microglia is activated in response to injury, infection, or other stress forms and produces proinflammatory cytokines, chemokines, and reactive oxygen species (ROS) that can damage neurons and disrupt normal brain function. An in vitro study using the microglial cell line BV2 observed that insulin reduced the expression of iNOS and TNF-α, which belong to proinflammatory markers of the M1 microglia subtype [28]. In line with this, a prior study in the human glial cell also suggested that insulin can increase the release of the anti-inflammatory marker from the M2 subtype of microglia, IL-8, while decreasing the production of monocyte chemoattractant protein-1 (MCP-1), hence reducing the cytotoxic effect of activated microglia [29]. Researchers have also investigated the interaction between IGF-1 and both subtypes of microglia. IGF-1 upregulation is associated with increased M2 phenotypes, such as induction of IL-4 or IL-13 [19,30]. Moreover, IGF-1 may inhibit the expression of TNF-α, a marker of the M1 phenotype [31].

Insulin and IGF-1 are essential for the survival and growth of neurons in the brain. Studies have shown that insulin and IGF-1 activate the PI3K/Akt signaling pathway, which plays a vital role in regulating cell survival and growth, as well as intracellular glucose uptake [32]. This pathway mediates the effects of insulin and IGF-1 on neuronal survival by activating anti-apoptotic pathways and inhibiting the expression of pro-apoptotic proteins, ultimately promoting neuronal survival [33]. In addition, insulin and IGF-1 can stimulate the brain-derived neurotrophic factor (BDNF), a protein playing a role in the survival and growth of neurons [34]. BDNF plays a crucial role in synaptic plasticity, neurogenesis, and neuronal survival in the brain. BDNF is also involved in several cognitive processes, such as long-term potentiation (LTP), learning, and memory [35]. IGF-1 has been shown to regulate the activity of signaling pathways that play a crucial role in the initiation and maintenance of LTP. Upon activating the IGF-1 receptor (IGF-1R), IGF-1 triggers the Ras/ErK signaling pathway, which in turn regulates the phosphorylation of calcium/calmodulin-dependent kinase 2 alpha (CaMKIIα), a regulator of synaptic plasticity. In an animal study, reduced levels of CaMKIIα phosphorylation and increased levels of MAPK/ErK phosphorylation were observed in the hippocampus of aged mice, which were associated with the upregulation of calcium-activated potassium channels. This led to the abolishment of synaptic potentials and a decline in memory function [36]. While the role of insulin in neuronal survival and growth is less well understood than that of IGF-1, evidence suggests that insulin may also play a role in these processes. A study using a hippocampal-specific insulin resistance model found that spatial learning and synaptic plasticity impairments were associated with the downregulation of IR in the hippocampus. Specifically, the downregulation of IR expression may lead to the disruption of LTP. These findings suggest that insulin may contribute to the regulation of neuronal plasticity [37]. Interestingly, a study published during the 2000s established that insulin might induce the release of soluble APPα fragments through PI3K activation in SH-SY5Y cells [38]. Soluble APPα is hypothesized to have a neuroprotective effect by modulating neuronal excitability, synaptic plasticity, neurite outgrowth, synaptogenesis, and cell survival [39].

Combined, insulin and insulin-like growth factor 1 (IGF-1) exert numerous neuroprotective effects in the brain, including regulation of glucose metabolism, promoting neuronal survival and growth, and inhibiting neuroinflammation. These effects are typically coordinated and occur concomitantly with the expression of insulin and IGF-1. Therefore, understanding the underlying mechanisms of insulin and IGF-1 in the brain is essential for the development of new treatments for neurodegenerative diseases.

### 2.3. Impaired Insulin Signaling in the Brain

In a normal state, insulin and IGF-1 exert their activities through two receptor tyrosine kinases, IR and IGF-1R, that are abundantly expressed in the brain. According to rodent studies, the olfactory bulb has the most significant expression of IR, followed by the cortex, hippocampus, hypothalamus, and cerebellum, with comparatively low levels in the striatum, thalamus, midbrain, and brainstem. IGF-1R is found to be abundant in the cortex, hippocampus, and thalamus, with moderate expression in the olfactory bulb, hypothalamus, and cerebellum and the least in the striatum, midbrain, and brainstem [40,41]. Previous studies on neurodegenerative diseases, mainly AD, have identified impaired insulin signaling as a crucial factor in AD. A study conducted in human postmortem brain tissue observed an impairment in insulin and IGF signaling. The reduced response of insulin signaling occurred mainly in the hippocampus and, to a lesser extent, in the cerebellar cortex. More importantly, the reduced responses occurred in the absence of diabetes [42]. In line with these findings, a study in aged APP/PS1 mice models has observed elevated IRS-1 pSer^616^, which is correlated with the impairment of cognitive function. Interestingly, peripheral insulin sensitivity is not affected even with the sign of insulin resistance in the brain. The presence of high levels of IRS in AD mice models suggests that Aβ accumulation in the brain is probably linked to impaired insulin signaling in the brain, regardless of peripheral insulin sensitivity [43]. Brain insulin resistance that is featured by decreased PI3K-Akt activity is also reported to trigger the phosphorylation of Tau by the disinhibition of GSK3β activities and the suppression of Tau phosphatases [44,45]. An animal study on AβPP/PS1 transgenic mice found that dysfunction in PI3K-Akt and MAPK/ERK signaling pathways occurs in a brain-specific and structure-specific manner in AD. Specifically, the PI3K-Akt cascade was dysregulated, with increased levels of inflammation-associated proteins such as NFκB and structure-specific regulation of proteins associated with autophagy. These findings suggest that impaired brain insulin signaling and its downstream signaling pathways, including PI3K-Akt and MAPK/ERK, may contribute to the pathogenesis of AD [46]. Generally, insulin resistance may contribute to AD pathogenesis through four mechanisms: dysregulation of production and clearance of Aβ, altering GSK-3β activity and promoting hyperphosphorylation of Tau, regulating microglial activation that leads to the exacerbation of neuroinflammation, and the disintegrated of tight junctions within the BBB as well as JNK-pIRS-1 pathway interruption. These mechanisms will be discussed further in the next section.

## 3. Involvement of Brain Insulin/IGF-1 Signaling in Dementia Pathologies

### 3.1. Alzheimer’s Diseases

AD continues to be the most common cause of dementia; hence, its pathologies are the most thoroughly studied. The well-known pathogenic hallmarks of AD are the accumulation of Aβ and Tau hyperphosphorylation. However, impaired insulin signaling in the brain has emerged as a significant contributor to AD pathogenesis. It has been hypothesized that the dysfunction of insulin signaling in the brain may be one of the mechanisms that trigger the onset and progression of AD, leading some researchers to refer to AD as “type 3 diabetes mellitus” due to the parallels between the impaired brain insulin signaling observed in AD and the insulin resistance observed in T2DM [47].

First, the presence of a toxic form of Aβ is determined by the balance between its production and clearance. This process may be controlled by processing Aβ precursor protein (APP) and insulin-degrading enzyme (IDE) activity. Insulin signaling is thought to play a role in regulating both the production and clearance of Aβ [48]. Insulin resistance has been observed to promote Aβ generation in the brain, which can occur due to changes in insulin signal transduction, modulation of β-secretase and γ-secretase activities, and the accumulation of autophagosomes in an animal study using mice brains [49]. Another in vivo study using streptozotocin (STZ)-treated APP/PS1 transgenic mice demonstrated a stimulated downstream of insulin signaling, GSK3, and c-Jun N-terminal kinase (JNK) signaling pathway. This stimulation leads to the activation of the amyloidogenic APP processing pathway, increasing the accumulation of Aβ [50]. On the other hand, Aβ clearance by IDE should also be considered. It is evident that Aβ shares similarities with insulin in several aspects. In fact, Aβ demonstrates direct competition with insulin for binding to the IR and IDE due to the resemblance of the sequence recognition motif [51]. IDE has a lower affinity for Aβ than insulin. Thus, it preferentially degrades insulin over Aβ when both are present in the brain. However, when insulin levels are high, as in the case of T2DM, IDE may become saturated with insulin and unable to degrade Aβ properly [52,53]. In addition, the previously mentioned postmortem study also suggested that the increase in Aβ oligomer may play a direct role in brain insulin resistance as it can inhibit the activation of IR and reduce its downstream signaling, such as IRS-1 for insulin signaling and IRS-2 for IGF-1 signaling, Akt pS^473^, GSK-3β, and mTOR pS^2448^ [42]. Taken together, an increase in the total number of Aβ due to production-clearance imbalance will eventually disrupt the insulin signaling that is already impaired, making another vicious cycle. This view aligns with a recent meta-analysis of IDE characteristics in AD patients. The result showed a significant decline in the IDE protein level of AD patients compared to the controls, specifically in the cortex and hippocampus [54]. It should be noted that this meta-analysis includes a limited number of studies due to insufficient literature. Nonetheless, the result might give a clue for further investigation about IDE.

Second, the regulation of brain insulin/IGF-1 signaling may contribute to tau phosphorylation. Studies have shown that impaired insulin and IGF-1 signaling pathways in AD lead to increased GSK-3β activity, resulting in hyperphosphorylation of Tau [55]. In a previous study, we reported that the brain of young senescence-accelerated mouse prone 8 (SAMP8) exhibits compensatory upregulation of inhibitory phosphorylation of GSK-3β, which corresponds with the inhibition of phosphorylation of Tau at Ser396. Although our results focused primarily on AMPK signaling, we noted this relationship between GSK-3β and Tau phosphorylation [56]. Insulin resistance in the brain reduces Akt function, resulting in GSK3 activation [57]. A study conducted both in vitro and in vivo has shown evidence that caspase-3 may promote the cleavage of Akt, which modulates the activation of the GSK3 pathway and eventually triggers Tau hyperphosphorylation [58]. Another possible mechanism is through the modulation of protein phosphatase 2A (PP2A), an enzyme that dephosphorylates Tau. Insulin signaling has been shown to increase PP2A activity, leading to decreased Tau phosphorylation. However, insulin resistance may reduce the effectiveness of insulin in activating PP2A in AD, leading to increased Tau phosphorylation [56,59,60]. A study using rodents injected with STZ observed that in an insulin-deficient state, the inhibition of PP2A was associated with Tau hyperphosphorylation. Additionally, they evaluated a decrease in Tau phosphorylation to the basal level after the injection of insulin 30 min before sacrificing the rodents [45]. Likewise, another AD animal model study also showed a similar result, highlighting the increase in phosphorylated Tau, as well as a decrease in PP2A levels in the hippocampus of STZ-injected groups [61].

Third, impaired brain insulin/IGF-1 signaling can result in inflammatory microglial activation and exacerbated neuroinflammation in AD. Insulin/IGF-1 signaling plays a crucial role in regulating microglial activation, leading to the inhibition of proinflammatory cytokines and the secretion of several anti-inflammatory cytokines, as shown in preclinical studies mentioned in the previous section [19,28,29,30,31,62]. Consequently, impaired insulin signaling can result in increased production of proinflammatory cytokines, such as IL-1β and TNF-α, which can exacerbate Aβ and Tau pathology and lead to another vicious cycle [63,64]. Additionally, astrogliosis has been known for paving the way for AD worsening by secreting proinflammatory cytokines [65]. Impaired brain insulin signaling can also affect microglial phagocytic activity, which is essential for clearing Aβ and other debris from the brain. Insulin signaling has been shown to enhance microglial phagocytosis via the triggering receptor expressed on myeloid cells 2 (TREM2)-PI3K-Akt signaling. The defect in this particular signaling may cause a decrease in Aβ clearance [66]. A recent study in AD mice showed that the deficiency of TREM2 in microglia could impair Akt-mTOR signaling and affect autophagy and energy metabolism [67].

Finally, the BBB theory is another important concept worth investigating when discussing AD. It is widely understood that BBB disruption caused by diabetes can impact how glucose and insulin are transported into neurons and glia, resulting in cognitive impairment and, consequently, AD development [68,69]. A few studies have focused on insulin resistance in the brain and its correlation with BBB dysfunction. Recent research perspectives have pinpointed the relation of brain insulin resistance with BBB through the expression of several proteins. Since peripheral insulin that crosses the BBB is the primary source of insulin in the brain, brain insulin resistance is strongly tied to peripheral insulin in terms of BBB alteration. When insulin resistance is present, the expression of proteins named high-mobility group box-1 (HMGB1) and receptor for advanced glycation end-products (RAGE) is elevated, resulting in the subsequent activation of toll-like receptor 4 (TLR4). This can result in cell death of brain endothelial cells, astrocytes, and pericytes. As a result, the integrity of tight junctions within the BBB is weakened. Additionally, the JNK pathway may also be activated by HMGB1, causing further impairment of brain insulin signaling through the inhibitory serine phosphorylation of IRS-1. Hence, there will be a subsequent loop of insulin resistance that affects the pathological features of AD, as mentioned above [70].

### 3.2. Vascular Dementia

Following AD, VaD is the second most prevalent type of dementia, and vascular pathology is frequently mixed with AD’s pathologies. According to the National Institute of Neurological Disorders and Stroke and the Association Internationale pour la Recherche et l’Enseignement en Neurosciences (NINDS-AIREN) criteria, VaD is diagnosed based on clinical and radiological evidence of cognitive decline and vascular factors related to the onset of dementia, including history of stroke, focal neurological signs or symptoms, or evidence of cerebrovascular disease on neuroimaging, indicating that ischemic damage is the primary cause of VaD [71]. Despite the fact that other risk factors can also contribute to VaD, many researchers have linked it to post-stroke cognitive impairment [72]. It is common to observe hyperglycemic conditions after the acute state of stroke. Following an acute state of stroke, hyperglycemia is frequently seen. This phenomenon has demonstrated the tight connection between vascular issues and insulin signaling. However, insulin signaling impairment in the brain requires prolonged cerebral hypoperfusion. It has been proposed that one of the reasons for insulin signaling disturbances is long-term cerebral hypoperfusion. In a 2013 animal study, researchers discovered that stroke model rats had a lower basal rate of phosphorylated IRS-1 at tyrosine residues, lower phosphorylated Akt, and higher phosphorylated AMP-activated protein kinase (AMPK) levels [73,74]. Moreover, IGF-1 mRNA and phosphorylated Akt levels have significantly decreased in a study using VaD model rats [75].

In addition to the animal study, the link between VaD and brain insulin resistance was also observed in the postmortem brain. This research discovered an increase in phosphorylated serine616 IRS-1 in the brain of deceased patients with cerebrovascular diseases. Interestingly, this finding was dominated by individuals without diabetes, demonstrating that VaD might be the only contributing factor [76]. It is crucial to remember that insulin signaling disruption directly affects the formation of Aβ, suggesting that the pathogenesis of VaD and AD may be overlapped. Recently, scientists have suggested using the IGF-1 level as a marker to differentiate VaD and AD. There has been a significant decrease in IGF-1 serum in VaD patients from a memory clinic, which is not observed in AD patients. Mixed VaD and AD dementia were classified as VaD in this study. Therefore, the IGF-1 level can also be used to identify whether or not VaD occurs in AD patients [77]. Nevertheless, valid biomarkers for differentiating VaD from AD have not been developed. Mixed VaD and AD pathologies are significant, indicating heterogeneity in pathogenic mechanisms of VaD, including insulin resistance.

### 3.3. Lewy Body Dementia

LBD is the third most recognized type of dementia. The term LBD refers to dementia with Lewy bodies (DLB) and Parkinson’s disease dementia (PDD). The neuropathology of this type of dementia is characterized by the aggregation of overexpressed α-synuclein (α-syn) oligomers into Lewy bodies in the cytoplasm of neurons. The link between LBD and insulin/IGF resistance in the brain is largely unexplored. Previous studies that employed postmortem human brain samples have demonstrated reduced binding of IGF-1 and IGF-2 receptors but comparable binding of insulin in the DLB brain, accompanied by the downregulation of insulin and IGF receptors. Similarly, the binding levels of IGF-1 and IGF-2 receptors in the PDD brain were lower than those in controls, although the difference was not statistically significant [78]. By these results, a recent study has also demonstrated the close link between IGF signaling and PDD. Lower plasma levels of IGF-1 are associated with worse cognitive function in Parkinson’s disease (PD) patients [79]. Recent longitudinal clinical studies evaluating CSF biomarkers and the progression of cognitive decline in PD patients demonstrated that PD patients with Aβ pathologic features in CSF showed more rapid progression of cognitive decline than PD patients without an average CSF profile [80,81]. Therefore, it should be noted that AD pathologic features are frequently observed in the brain of PDD and DLB patients, indicating that insulin or IGF resistance induced by pathologic proteins, such as Aβ and α-syn, may be a common feature of neurodegenerative pathogenesis. Based on the data discussed above, it may be hypothesized that IGF signaling is more relevant to LBD than insulin signaling. However, further studies that explore the association of insulin or IGF resistance with proteinopathies in LBD and AD are required.

### 3.4. Frontotemporal Dementia

The next type of dementia that will be discussed is FTD. This particular form of dementia is caused by the specific degeneration of gray and white matter strictures in the frontal and temporal lobes, in which abnormal deposits of Tau or TDP-43 proteins and activation of glial cells induce neurodegeneration. Compared with AD, FTD patients show progressive abnormalities in behavior, language, or personality, as well as cognitive decline [82,83]. Peripheral insulin resistance and FTD have been linked in earlier studies. The HOMA-IR index of FTD, AD, and healthy subjects differs noticeably, according to a study from the early 2010s. In contrast to AD patients, FTD patients had a HOMA-IR index that was approximately twice as high as healthy subjects [84]. However, it is not clear whether hyperinsulinemia and insulin resistance are detrimental to FTD progression. In addition, the presence of brain insulin resistance or any defects in the insulin signaling pathway in the FTD brain remains to be clarified. Even so, many opportunities for further research can be developed with those findings. Consistent with this result, another study using postmortem FTD brain has demonstrated the alterations of insulin and IGF signaling. IGF-1 level was elevated in the frontal lobe, along with upregulation of IGF-1R in most frontal and temporal areas. Meanwhile, IGF-2R was downregulated in frontal Brodmann’s area 24 and temporal Brodmann’s area 38. Insulin and IR expression were also elevated in the frontal lobe. Together, insulin signaling may be involved in the pathogenesis of FTD [83]. Nevertheless, the insulin resistance observed in FTD patients is not specific to the disease but overlaps with AD and other types of dementia. Therefore, further studies on the association of insulin resistance with pathologic features of FTD, such as TDP-43 or tau accumulation, are necessary.

## 4. Preclinical Evidence of Anti-Diabetic Drugs for Alzheimer’s Drug Repositioning

Given that impaired insulin signaling in the brain is one of the characteristics of AD patients, strategies for repurposing anti-diabetic drugs as a therapeutic option for dementia have been proposed (Figure 1). The most studied candidates for drug repositioning of anti-diabetics are insulin and insulin sensitizers, although several anti-diabetic drugs also have been tested. Overall, preclinical evidence has shown favorable pharmacological effects that support the idea to repurpose anti-diabetic drugs for dementia (Table 1).

### 4.1. Insulin

Insulin has been administered subcutaneously for the treatment of T2DM. However, the subcutaneous route has several limitations for dementia patients, including the discomfort of frequent injections and the probability of hypoglycemic events. To resolve the limitations, recent investigations have concentrated on the intranasal route as a potential substitute since insulin receptor expression is abundant in olfactory bulbs. Intranasal insulin is also anticipated to offer an increased probability of patient compliance and a lower risk of hypoglycemia than parenteral insulin [85,86]. Preclinical and clinical studies have reported positive results regarding the use of intranasal insulin in cognitive decline, notably AD, although not conclusive yet. A study in animals highlighted the superiority of intranasal insulin over parenteral insulin as a treatment for cognitive decline. Insulin was delivered to all regions of the mouse brain when supplied intranasally but not when administered intravenously. In addition, the amount of intranasally administered insulin detected in mice serum is insufficient to induce a hypoglycemic state [86]. In terms of Aβ deposition, intranasal insulin reduces the amount of Aβ production and plaque formation after six weeks of treatment in APP/PS1 mice, which is mostly caused by the ability of intranasal insulin to reduce BACE1 protein levels. Additionally, this study observed an improvement in brain insulin signaling following intranasal insulin treatment. Specifically, the treatment resulted in a significant decrease in the phosphorylation of IRS1 at S^612^ and S^636^ and the restoration of impaired levels of IR and Akt. However, the potential direct association between the amelioration of brain insulin signaling impairment and the decrease in BACE1 levels has yet to be confirmed [87]. In a recent study, intranasal insulin treatment ameliorated the Aβ oligomer-induced cognitive impairment in male rats when tested with the Morris water maze [88]. This result also aligns with another animal study that compared the effect of intranasal insulin and subcutaneous insulin in brain insulin signaling. Intranasal insulin treatment for four weeks in the rat AD model was able to reduce the hyperphosphorylation of Tau in the hippocampus and cerebral cortex more significantly than the conventional subcutaneous insulin administration. The treatment of insulin through intranasal administration has proven to attenuate the overactivation of GSK3β as well as the reduced phosphorylation of Akt [89]. Meanwhile, several preclinical studies have suggested that the administration of insulin can ameliorate the activation of microglia or astrogliosis. A study using AD model mice found that intranasal insulin treatment reduced the microglia marker, the mouse homolog of human cluster of differentiation 68 (CD68). This finding confirms the role of impaired insulin signaling in the activation of microglia. The researchers later revisited this study but with long-term intranasal insulin treatment. They discovered that six weeks of intranasal insulin treatment in vivo could reduce microglial activation [90,91]. In another paper, the researchers also suggested that astrogliosis in rat AD models can be restored by six weeks of intranasal insulin administration [92].

### 4.2. Insulin Sensitizer

Several studies have investigated the potential of metformin as an AD treatment. However, the results have been inconsistent and inconclusive. Previous reviews have described the beneficial role of metformin in AD through several mechanisms, including the reduction in Aβ plaques and neurofibrillary tangles formation. In addition, metformin has been shown to inhibit inflammation, normalize altered mitochondrial dysfunction and reduce oxidative stress, attenuate microglial overactivation, and promote neuronal survival mainly through the AMPK pathway. These mechanisms suggest that metformin could positively impact AD pathophysiology [93]. An animal study of intranasally administered metformin for four weeks showed the mechanism behind its effect of ameliorating cognitive decline in the AD mice models through brain insulin signaling. The intranasal administration of metformin was shown to increase the sensitivity of insulin signaling in hippocampal and cortical tissue by increasing the phosphorylation of IR and Akt [94]. In line with these results, an 18-week treatment of db/db mice with metformin preserved the neuronal structures by attenuating the decrease in the synaptic protein and attenuating the hyperphosphorylation of Tau [95]. At the amyloidogenesis level, the evidence from preclinical studies is contradictive. One study suggested that metformin exposure reduces Aβ accumulation by decreasing the protein level of BACE1, the main amyloidogenic enzyme [96]. On the contrary, another preclinical study observed an increase in BACE1 expression after metformin treatment, which might be due to AMPK activation, one of the main targets of metformin [97]. In addition, existing preclinical studies also have demonstrated the benefit of metformin in vascular-related dementia. Metformin-treated mice with cerebral amyloid angiopathy and T2DM significantly reduced Aβ deposits in cerebral blood vessels by increasing the IDE expression [98]. A population-based case–control study in the United Kingdom compared the risk of AD in users of metformin or other anti-diabetic drugs. Surprisingly, the study found that long-term consumption of metformin is associated with a slightly higher risk of developing AD. In comparison, other anti-diabetic drugs were not associated with higher AD risk. This result is not consistent with the findings of in vitro or animal studies, in which metformin showed potential benefits in reducing the risk of developing AD [99]. However, it should be noted that the diagnostic accuracy of AD by the general practitioner using clinical diagnostic criteria may not be high, since patients with VaD or dementia with mixed-type pathologies might be included. Meta-analyses of clinical observational studies also failed to demonstrate the beneficial effect of metformin on AD and neurodegenerative diseases in general. In fact, metformin might increase the risk of AD among Asians, as well as PD [100,101]. Although the negative results may be due to the limited number of samples or low accuracy of clinical diagnosis, not the biological diagnosis of AD, the utilization of metformin as a monotherapy drug for dementia should be assessed further.

Thiazolidinediones are another frequently prescribed insulin sensitizer that is now being studied for their impact on cognitive decline. As a peroxisome proliferator–activated receptor gamma (PPARγ) agonist, thiazolidinediones increase insulin sensitivity by binding into PPAR to facilitate glucose uptake in the peripheral organs and to reduce gluconeogenesis in the liver. Examples of drugs in this category are rosiglitazone and pioglitazone [102]. Currently, preclinical studies regarding thiazolidinediones’ effect on cognitive decline have shown favorable outcomes supporting the probability of drug repositioning. A study conducted on cultured primary hippocampal neuronal cells demonstrated the efficacy of thiazolidinediones, particularly pioglitazone, in facilitating the degradation of Aβ and diminishing its production. The findings revealed a significant increase in the expression of IDE and a decrease in the expression of BACE1 as well as APP. These alterations impacted both mRNA levels and protein expression [103]. Meanwhile, rosiglitazone has been observed to ameliorate memory deficits in transgenic mice overexpressing human amyloid precursor protein (hAPP), mostly due to its ability to promote Aβ clearance and reduce phosphorylated Tau aggregates [104]. In another animal study, Western blot and immunohistochemistry analyses using the hippocampus of DM rats demonstrated increased levels of IR, IRS-1, Akt, phosphorylated CREB, and Bcl-2 compared to the control groups. Administration of rosiglitazone led to normalization of these markers, although the observed decrease did not reach statistical significance. In addition, DM rats treated with rosiglitazone were noticeably superior during the water maze test compared to the untreated DM rats, indicating an association of rosiglitazone-mediated improvement of insulin signaling with cognitive improvements [105]. Interestingly, another study found an association between increased Aβ clearance and low-dose rosiglitazone treatment in the primary human brain microvascular endothelial cells. This association can be attributed to the capacity of rosiglitazone to enhance the transcription and translation of low-density lipoprotein receptor-related protein 1 (LRP1), which has been suggested to facilitate the efflux of Aβ across the BBB [106]. Beneficial effects of thiazolidinediones were also seen in STZ-induced diabetic rats with vascular dementia. Administration of rosiglitazone and pioglitazone improved not only learning and memory ability but also endothelial function. Moreover, the researchers compared the results between thiazolidinediones-treated rats and donepezil-treated rats. Donepezil has been used widely in the clinical setting to reduce the symptoms of AD patients. Remarkably, the data revealed that the two groups had comparable outcomes [107]. Furthermore, population-based studies suggest that thiazolidinediones may be a more effective option than metformin for reducing dementia incidence in the Asian population. A recent cohort analysis in the Chinese population observed a 49% reduction in dementia incidence when treated with thiazolidinediones [108]. Despite some positive findings in studies, clinical trials investigating PPARγ agonists have not shown promising results. Many trials for rosiglitazone and pioglitazone did not observe favorable outcomes and were either terminated early in phase III or failed to show good results (see the next section). Therefore, further studies are necessary to fully comprehend the potential benefits and risks of thiazolidinediones for treating cognitive decline.

### 4.3. Insulin Secretagogues

Glucagon-like peptide-1 (GLP-1) receptor agonists are among the most promising candidates for dementia drug repositioning. Activation of the GLP-1 receptor agonist has been shown to increase cyclic AMP (cAMP) levels, which in turn leads to the activation of protein kinase A (PKA) and other downstream signaling pathways. These pathways regulate synaptic plasticity, neuronal survival, and glucose metabolism, all of which are important for maintaining cognitive function. GLP-1 receptor activation has also been shown to improve glucose uptake and utilization in the brain, which may help to protect against neuronal damage and cognitive decline [109]. Preclinical studies have provided evidence for its potential in the treatment of dementia. For instance, a study utilizing double transgenic tau and amyloid precursor protein (TAPP) mice found that treatment with the GLP-1 receptor agonist exendin-4 reduced amyloid plaque deposition and phosphorylated Tau. This study also emphasized that the aforementioned effects of exendin-4 were prominent in an insulin-deficient condition, rather than insulin resistance [110]. Similarly, another study found that another GLP-1 receptor agonist, liraglutide, can improve cognitive function, reduce neuronal loss, and decrease vascular damage in mice with a mixed model of AD and T2DM. These results suggest that liraglutide may be superior for the treatment of cognitive decline in individuals with coexisting AD and T2DM. However, no discernible differences were detected in IR or IGF-1 mRNA levels of the control and treatment groups, raising intriguing questions about whether a direct correlation between the advantageous effects of liraglutide and brain insulin signaling existed or not [111]. In addition, combining GLP-1 receptor agonist with insulin led to improved memory performance of AD model mice compared to control AD. Additionally, gene expression analysis showed that the combination treatment restored the expression of genes involved in insulin receptor pathways that were downregulated in the brain of AD mice, accompanied by the neuroprotective effects [112]. Nonetheless, liraglutide contributed to the decrease in Aβ accumulation by increasing IDE levels in APP/PS1 mice, which leads to improved Aβ clearance [113]. In a study of nonhuman primates, liraglutide offered partial protection against the loss of receptors and synapses in the frontal cortex, hippocampus, and amygdala [114]. Additionally, it is postulated that liraglutide exerts a mitigating effect on hemorrhage burdens, as indicated in the aforementioned study, although the exact mechanism is not yet elucidated [111]. Furthermore, a study analyzing data from eight randomized controlled trials as well as nationwide disease and prescription registers showed that the use of GLP-1 receptor agonists was associated with a reduced risk of dementia [115].

Sulfonylurea is another anti-diabetic drug classified as an insulin secretagogue. Unlike GLP-1 receptor agonists, there is limited evidence supporting sulfonylurea as a treatment for AD and other dementias. Most of the preclinical studies have investigated its anti-inflammatory and anti-Aβ effects in AD. One study evaluated the effects of glibenclamide, a sulfonylurea, on microglial activation and Aβ deposition in 5XFAD mice. The results showed that glibenclamide inhibited the deposition of Aβ in the hippocampus of the mice. Furthermore, the study showed that glibenclamide treatment inhibited microglial activation in vitro and in vivo. This suggests that the therapeutic effect of glibenclamide on AD may be due to its ability to modulate the microglial function and reduce neuroinflammation [116]. Likewise, glibenclamide is shown to improve memory impairment and reduce hippocampal inflammation in rats with type 2 diabetes and sporadic AD [117]. Intriguingly, another study suggested that glibenclamide attenuated depression and anxiety-related symptoms in a rat model of AD by regulating the hypothalamic–pituitary–adrenal axis. However, the specificity of the observed effect in AD models remains uncertain [118]. In contrast with preclinical studies, sulfonylurea appeared to be less beneficial compared to metformin in a retrospective cohort study. Patients treated with metformin had a lower incidence of AD and VaD compared to those treated with sulfonylureas [119]. Therefore, while sulfonylurea has shown some potential benefits in reducing several dementia pathologies in preclinical studies, it has limited evidence in human studies. Further studies to elucidate the pharmacological mechanisms are needed to fully understand the therapeutic effects on cognitive impairment.

### 4.4. Other Anti-Diabetic Drugs

Several anti-diabetic drugs that are not classified as insulin sensitizers or insulin secretagogues are also under investigation for drug repositioning. These include sodium/glucose cotransporter 2 (SGLT-2) inhibitors and amylin analogs. SGLT-2 inhibitors such as canagliflozin, dapagliflozin, empagliflozin, and ertugliflozin reduce the reabsorption of glucose in the kidneys. These SGLT-2 inhibitors have a favorable safety profile as their mechanisms of action are unrelated to pancreatic beta cell function or insulin signaling. Furthermore, SGLT-2 inhibitors exhibit anti-inflammatory properties, potentially contributing to neuroprotective effects and metabolic benefits [120,121]. The link between SGLT-2 inhibitors and dementia is still a topic of active research; however, there is currently no definitive evidence supporting the therapeutic efficacy of SGLT-2 inhibitors on dementia. In a recent animal study, SGLT-2 inhibitors were found to ameliorate the cognitive impairment associated with diabetes. In this research, SGLT-2 inhibitors restored neurotrophin levels in the brain. Neurotrophins are proteins that support the growth and survival of neurons, and their depletion has been linked to cognitive dysfunction. By increasing the levels of these important proteins, SGLT-2 inhibitors can contribute to the improvement of cognitive function. Additionally, SGLT-2 inhibitors have been shown to reduce the levels of inflammatory cytokines in the brain of diabetic mice [120]. Meanwhile, another study focused on the autophagic machinery of SGLT-2 inhibitors in the AD rat model. In this study, dapagliflozin was able to enhance autophagy via the LKB1/AMPK/SIRT1 pathway, which has been shown to be disrupted in AD patients [122]. These results are in line with a nationwide population-based longitudinal cohort study based in Taiwan. The study included over 150,000 patients with type 2 diabetes who were treated with SGLT2 inhibitors or other diabetes medications. In this study, the use of SGLT2 inhibitors was associated with a significantly lower risk of incident dementia compared to the use of other diabetes medications [121]. Further research is needed to confirm these findings and to determine whether this drug could be effective in humans.

Several studies are currently under investigation to clarify the impact of amylin analogs on dementia. Amylin analogs are synthetic versions of amylin, a hormone that is produced by pancreatic beta cells. Amylin has been reported to interact with Aβ in several ways, although not yet fully elucidated. First, amylin is thought to inhibit the aggregation of Aβ into plaques, which later helps to reduce the accumulation of Aβ in the brain. Second, amylin may enhance the clearance of amyloid-beta from the brain by promoting its uptake and transport across the BBB. Finally, amylin could possibly also bind directly to amyloid-beta and prevent its toxicity to neurons [123,124]. Pramlintide, one of the amylin analogs, has demonstrated potential neuroprotective effects in a mouse model of AD. The key finding of the study was that pramlintide treatment resulted in reduced accumulation of Aβ in the brain. The study also found that pramlintide treatment improved cognitive function and reduced oxidative stress and neuroinflammation in the brain, both of which are implicated in AD pathogenesis [125]. In a STZ-induced rat model of AD, pramlintide demonstrated superior performance in certain memory tests when compared to metformin [126]. Nevertheless, the use of amylin analogs as a DMT in dementia should be assessed further, especially when considering the similar ability of amylin and Aβ to be aggregated and misfolded [127].

**Table 1 ijms-24-11450-t001:** Summary of mechanisms of action and pharmacological effects of anti-diabetic drugs exhibiting neuroprotection in preclinical studies.

Classification	Drugs	Mechanisms of Action	Pharmacological Effects	References
Insulin	Intranasal insulin	- Decrease amyloidogenesis - Decrease neurofibrillary tangles formation - Decrease microglial overactivation	- Reduced cognitive impairment	[86,87,88,89,90,91,92]
Insulin Sensitizer	Metformin	- Decrease amyloidogenesis (contradictive) - Increase Aβ clearance - Decrease neurofibrillary tangles formation - Decrease microglial overactivation - Increase neuronal survival - Reduce mitochondria dysfunctions	- Reduced cognitive impairment - Reduced Aβ plaques - Reduced neuroinflammation	[94,95,96,97,98]
	Thiazolidinediones - Rosiglitazone - Pioglitazone	- Decrease amyloidogenesis - Increase Aβ clearance - Decrease neurofibrillary tangles formation - Preserve endothelial functions	- Reduced cognitive impairment - Improved endothelial function	[103,104,105,106,107]
Insulin secretagogues	Glucagon-like peptide-1 - Exendin-4 - Liraglutide	- Decrease amyloidogenesis - Increase Aβ clearance - Decrease neurofibrillary tangles formation - Preserve endothelial functions	- Reduced cognitive impairment - Reduced neuronal loss - Reduced vascular damage	[110,111,112,113,114]
	Sulfonylurea - Glibenclamide	- Decrease microglial overactivation	- Reduced memory impairment - Reduced neuroinflammation - Reduced depression and anxiety-related symptoms	[116,117,118]
Other anti-diabetics	Sodium-glucose cotransporter-2 - Dapagliflozin - Empagliflozin - Canagliflozin	- Increase neuronal survival - Increase autophagy	- Reduced cognitive impairment	[120,122]
	Amylin analogs - Pramlintide	- Increase Aβ clearance	- Reduced cognitive impairment - Reduced Aβ plaques - Reduced neuroinflammation	[125,126]

## 5. Current Status of Clinical Trials for Anti-Diabetic Repositioning as Anti-Dementia Drugs

To date, the majority of preclinical and observational research findings have been encouraging to repurpose anti-diabetic medications as anti-dementia treatments. Despite mixed outcomes, anti-diabetic medication does reduce the incidence of dementia compared to no anti-diabetic treatment at all [128]. Regardless of how intriguing and feasible it is to repurpose anti-diabetic medications for dementia, none of the clinical trials have been successful. Given the lack of success in clinical trials thus far, we sought to investigate the scope and outcomes of previous and ongoing clinical trials related to the repositioning of anti-diabetic drugs as possible treatments for dementia. Using the clinicaltrial.gov database, we conducted a comprehensive search for clinical trials involving the drug candidates. We identified relevant trials using specific keywords related to the condition or disease, including “Dementia”, “Alzheimer’s disease”, “Vascular dementia”, “Frontotemporal dementia”, “Parkinson disease dementia”, “Lewy bodies dementia”, “Pick diseases”, “Cognitive impairment”, and “Cognitive decline”. In addition, we searched for interventions or treatments using keywords such as “Insulin”, “Biguanide”, “Metformin”, “Sulfonylurea”, “Insulin sensitizer”, “Insulin secretagogue”, “Rosiglitazone”, “Pioglitazone”, “Thiazolidinediones”, “GLP1”, “SGLT2”, “Amylin”, and “Pramlintide”. Our search was up to date as of 13 March 2023. The Boolean operator “OR” was used to combine keywords. With the aforementioned keywords, a total of 108 clinical trials were identified and manually screened to ascertain their relevance to our research interest. Trials that did not involve anti-diabetic drugs or targeted conditions that were not related to our research focus were excluded. After exclusion, a total of 60 clinical trials were found. Currently, many of the clinical trials are in phase II, which is a critical stage of drug development in which the efficacy and safety of the drug are evaluated in patients (Figure 2). Despite the considerable number of phase II clinical trials conducted on drugs for dementia, the progression rate of such drugs to phase III remains limited. Moreover, a majority of these clinical trials are focused primarily on AD, MCI, and general cognitive functions, while a relatively low number of clinical trials are directed toward other types of dementia (Figure 3). The most frequent focus of clinical trial is on insulin sensitizers, with insulin substrates being the second most studied group (Figure 4).

Intranasal administration of insulin allows for direct delivery to the brain, bypassing the peripheral effects of insulin on glucose regulation. Moreover, intranasal insulin has been shown to have a favorable safety profile compared to intravenous insulin, making it an attractive candidate for further study in the treatment of dementia. However, the clinical trials of intranasal insulin as a treatment of AD in humans are inconclusive. Intranasal insulin has been proven to improve cognitive function in healthy humans since the mid-2000s. In the previous study, the administration of intranasal insulin for eight weeks resulted in the improvement of delayed recall which depicted declarative memory performance, but not immediate recall. Thus, intranasal insulin is thought to improve intermediate to long-term memory [129,130]. This hypothesis was later supported by a pilot clinical trial in which patients with amnestic mild cognitive impairment (MCI) or mild to moderate AD were given insulin 20 IU or 40 IU intranasally (NCT00438568). The result of this phase II study showed promising treatment outcomes, as the administration of intranasal insulin improved delayed memory ability, Dementia Severity Rating Scale (DSRS) score, and Alzheimer Disease’s Assessment Scale—cognitive subscale (ADAS-cog) score [131]. In terms of treatment response, intranasal insulin was reported to have a variation between male and female participants as well as ApoE gene carriers. Male participants showed cognitive improvement for the 40 IU dose, while female participants showed functional preservation on either dose. Additionally, the study found that the gender difference was most apparent for individuals who did not carry the ApoE4 gene, with ApoE4-negative men improving and ApoE4-negative women worsening [132]. The research group subsequently conducted two additional clinical trials (NCT01547169, NCT01595646) to further elucidate the mechanism of action. The first trial investigated the impact of long-acting intranasal insulin detemir on cognitive function, while the second trial examined the effects of both regular and long-acting insulin on cognitive function and AD pathologies. Both studies found that insulin treatment was associated with improvements in cognitive function in patients with MCI or early-stage AD. Additionally, the second clinical trial found that insulin treatment was also associated with reductions in Aβ and tau abnormality in cerebrospinal fluid, supporting that insulin may have a disease-modifying effect on AD pathologies [133,134]. On the other hand, a more recent trial, which was meant to extend the previous findings in a longer and larger multisite randomized double-blinded clinical trial, could not observe any significant cognitive improvement from the participants diagnosed with amnestic MCI or AD after being treated with intranasal insulin for more than 12 months (NCT01767909). Nonetheless, this trial showed no adverse side effects after the treatment, assuring that intranasal insulin can be considered a safe therapy [135]. Meanwhile, the clinical trial regarding insulin treatment in other forms of dementia is limited. A clinical trial (NCT04115384) evaluating the efficacy of intranasal insulin in frontotemporal dementia was suspended due to the COVID-19 pandemic. It is currently unknown whether the trial will resume or not. Hence, while intranasal insulin has shown promising results in improving cognitive function and potentially modifying AD pathologies, clinical trials have generated mixed results. A personalized approach may be necessary for designing and conducting future trials, taking into account individual differences in sex and ApoE genotype. Despite these challenges, the favorable safety profile of intranasal insulin suggests that it may still be a viable therapy option for those with cognitive impairment.

Clinical trials evaluating insulin sensitizers as potential DMTs for dementia have yielded varying results based on the drug classes studied. Regarding metformin, individual clinical studies have generated unfavorable results. Two phase II trials (NCT01965756, NCT00620191) indicated no substantial enhancement in cognitive function with metformin administration [136,137]. Moreover, concerns have been raised about the potential side effects of metformin, including gastrointestinal symptoms [137] and vitamin B12 deficiency [138,139]. Another phase IV clinical trial was prematurely terminated due to financial constraints and the resignation of staff during the pandemic. Nevertheless, ongoing clinical trials investigating the effects of metformin on Alzheimer’s disease are currently enrolling participants, indicating that additional mechanisms may be elucidated in the future. For example, MetMemory (NCT04511416) and Multidomain Alzheimer Prevention with Targeted Metformin and Lifestyle Intervention (MET-FINGER, NCT05109169) are two out of several clinical trials that are currently in the early stages of investigating the effects of metformin on cognitive function and brain health in individuals with MCI or at risk of AD. Interestingly, an ongoing clinical trial is currently testing the safety of metformin in patients with C9orf72 repeat expansion (NCT04220021). C9orf72 repeat expansion is a genetic abnormality that has been strongly associated with inherited FTD. This genetic mutation is the most common cause of inherited FTD and is also associated with amyotrophic lateral sclerosis (ALS) [140]. Several clinical trials have also investigated the potential use of thiazolidinediones as medications for dementia. A phase III study named TOMMORROW (NCT01931566) showed no significant difference in cognitive outcomes between the pioglitazone and placebo groups. While there was a trend toward improvement in the pioglitazone group, this did not reach statistical significance. However, the study had limitations, such as early termination due to an unrealistic assumption of drug effect, a healthy and educated study population, investigation of only one drug dose, and a lack of baseline neuropathology biomarkers. These limitations increase the risk of falsely negative results [141]. In a phase II study (NCT00736996) examining the effects of pioglitazone or exercise on cognitive function in older adults with MCI and insulin resistance, improved insulin resistance was observed, but no significant improvement in cognitive performance was found [142]. Meanwhile, the clinical trial for rosiglitazone exhibited outcomes similar to the clinical trials conducted with pioglitazone. Despite promising results in a pilot study that demonstrated improvement in measures of delayed memory and selective attention in subjects with AD or amnestic MCI after six months of rosiglitazone treatment [143], phase III trials have failed to observe positive results [144,145]. The reported increased risk of heart failure associated with rosiglitazone has also raised concerns about its safety and presents challenges for the continuation of future clinical trials [144].

GLP-1 agonist is one of the possible candidates for drug repurposing from the insulin secretagogues category. However, relatively fewer clinical trials are currently investigating this class of drugs compared to others. Drugs in this category that have undergone trials are liraglutide and exenatide. In a clinical trial that has been completed (NCT01469351), it was observed that when liraglutide was administered for a period of 26 weeks, the expected decline in brain glucose metabolism was prevented. However, there were no significant differences in terms of cognition or amyloid deposition between the group that received liraglutide and the group that received a placebo [146]. A randomized controlled trial, registered under the name of ELAD (NCT01843075), is investigating the effects of liraglutide in individuals with Alzheimer’s disease. Although the status of the trial is currently unknown in the database, the primary endpoint aims to evaluate the change in the cerebral glucose metabolic rate in the cortical regions from baseline to follow-up in participants receiving liraglutide compared to those receiving a placebo. The study’s objective is to further evaluate the potential of liraglutide in improving cognitive function and reducing the progression of AD. By increasing the duration of the trial and enlarging the sample size, it is anticipated that the results of this study will produce more conclusive findings than the aforementioned trial [147]. A clinical trial of exenatide, intended primarily to assess its effectiveness in treating PD, also evaluated cognitive function as a secondary outcome (NCT01971242). This trial has been completed. The findings indicated that there were no significant differences in cognitive function between the exenatide and placebo groups. Subsequently, a new phase III clinical trial is currently in progress (NCT02847403). This trial is specifically designed to investigate the effects of exenatide on individuals with mild cognitive impairment. Remarkably, a phase II clinical trial conducted in high-probability AD patients found that exenatide did not produce any significant differences compared to placebo in clinical, cognitive, or biomarker outcomes, except for a reduction in Aβ42 in plasma neuronal extracellular vesicles. This reduction supports the idea that plasma neuronal extracellular vesicles may provide a platform for demonstrating biomarker responses in AD clinical trials, but the study was underpowered due to early termination. Thus, no firm conclusions can be drawn about exenatide’s potential as a disease-modifying treatment for AD from this study [148]. Therefore, although GLP-1 agonist has shown remarkable results in preclinical studies, their effects on cognitive function and amyloid deposition are still uncertain. Ongoing and upcoming clinical trials, such as the phase III trial investigating the effects of exenatide on individuals with mild cognitive impairment, might provide further insights into the therapeutic potential of GLP-1 agonists in treating dementia.

Scientists also have hypothesized that SGLT2 inhibitors may have potential benefits for dementia patients. Clinical trials are currently underway to investigate the effects of SGLT2 inhibitors on cognitive function and brain health in individuals with dementia. SGLT-2 inhibitors have acted as a comparator in several clinical trials involving anti-diabetic drugs for cognitive functions (NCT05313529, NCT03961659, NCT05081219). Following this, an ongoing pilot clinical trial of dapagliflozin as a DMT for AD is in progress (NCT03801642). The trial employs a randomized, double-blind, placebo-controlled design, with participants assigned to receive either dapagliflozin or placebo. The main outcome of this study is to investigate the effects of daily administration of 10 mg dapagliflozin for 12 weeks on n-acetyl aspartate (NAA) levels in individuals with probable AD. Alongside this, the study also evaluates the safety and tolerability of dapagliflozin as well as cognitive performance. The results of this study will provide useful insights into the potential therapeutic applications of dapagliflozin for individuals with AD, which could aid in the development of future drug repositioning approaches. Interestingly, a current clinical trial (NCT05565976) is actively recruiting participants to investigate the impact of dapagliflozin on cognitive impairment in stroke. Participants will receive 10 mg of dapagliflozin orally once a day for 12 months in addition to the standard stroke treatments. The primary outcome measure will be global cognitive function, while the secondary outcome will focus on the relationship between cardiovascular risk factors and cognitive decline. The trial is estimated to be completed by December 2024. The results of this study have the potential to provide valuable insights into the potential of dapagliflozin as a therapeutic agent for VaD following stroke.

Lastly, an emerging anti-diabetic drug that is worth discussing is the amylin analog. As explained in the previous section, it is evident that amylin analog is related to the reduction in Aβ accumulation, one of the main hallmarks of AD [125,126]. Pramlintide, an amylin analog and FDA-approved drug currently used for the treatment of diabetes, is now undergoing an early phase I clinical trial for AD drug repositioning (NCT03560960). In this multicenter clinical trial, the researchers aim to create a blood-based test for the early detection of AD by utilizing a single injection of pramlintide. If pramlintide is effective in releasing Aβ from the brain into the blood, this may also offer therapeutic potential by decreasing Aβ load.

It is important to note that the pathological features of dementia, for example, Aβ accumulation, occurred for decades before the symptoms started [149]. Thus, drugs targeting only specific stages of dementia may not be successful in late-phase clinical trials. Consequently, the study design of clinical trials may not have been designed to effectively test the efficacy of the anti-diabetic drugs being tested. For example, the dose or duration of treatment may not have been optimal. Adding another layer of complexity, dementia is a complex and heterogeneous disease, and different patients may have different underlying causes and disease mechanisms. This could make it difficult to identify a drug or treatment with a single mechanism that is effective for all patients with dementia. Furthermore, the enrolled patients in clinical trials for DMT based on clinical diagnosis might be biologically heterogeneous. The accuracy of AD is less accurate compared to biological diagnosis, specifically through the identification of amyloid or tau positivity. Current biological diagnosis of AD based on amyloid or tau positivity is widely acceptable to differentiate patients into specific pathological features [150]. According to the NIA-AA Research Framework, the definition of AD in a living person should be seen from a syndromal to a biological construct. Biomarkers determined by biochemical biomarkers (e.g., CSF Aβ or phosphorylated Tau level) or imaging biomarkers are grouped into evidence of ‘A’ (Aβ deposition), ‘T’ (pathologic tau), and ‘N’ (neurodegeneration). This A/T/(N) classification will enable a more accurate characterization that is associated with AD, which might enable enrolling the patients more homogeneously in clinical trial design. The enrollment of homogeneous patients in clinical trials may increase the power of statistics when efficacy and safety are evaluated. Therefore, future clinical trials using the ‘A/T/(N)’ biologic classification for anti-diabetic trials may contribute to successful drug repositioning and the understanding of the pharmacological mechanisms of action. Overall, the reasons behind the inconclusive results of trials related to anti-diabetic drug repositioning as a DMT for dementia are likely multifactorial. Further research is needed to better understand the underlying causes of dementia and to develop effective treatments.

## Figures and Tables

**Figure 1 ijms-24-11450-f001:**
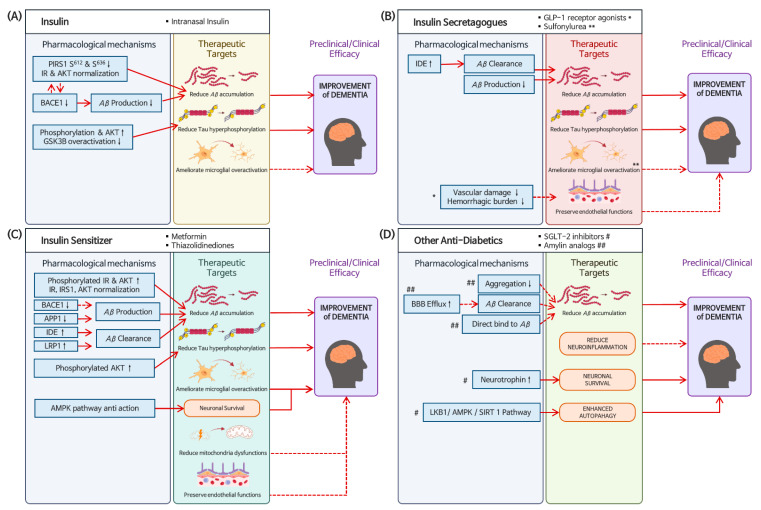
Current proposed mechanisms of anti-diabetic drugs, including (**A**) insulin, (**B**) insulin secretagogues, (**C**) insulin sensitizers, and (**D**) SGLT-2 inhibitors and amylin analogues as DMT for dementia. * GLP-1 receptor agonists, ** Sulfonylurea, # SGLT-2 inhibitors, ## Amylin analogs, solid red line: current proposed mechanism, dashed red line: further studies needed.

**Figure 2 ijms-24-11450-f002:**
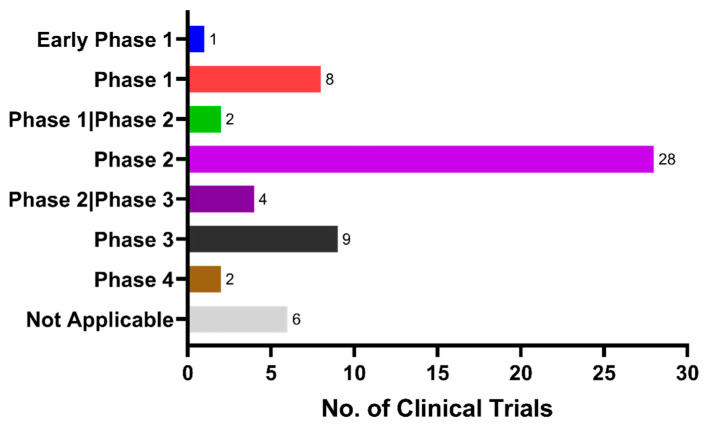
Current status of clinical trial phases in repositioning anti-diabetic drugs for dementia treatment.

**Figure 3 ijms-24-11450-f003:**
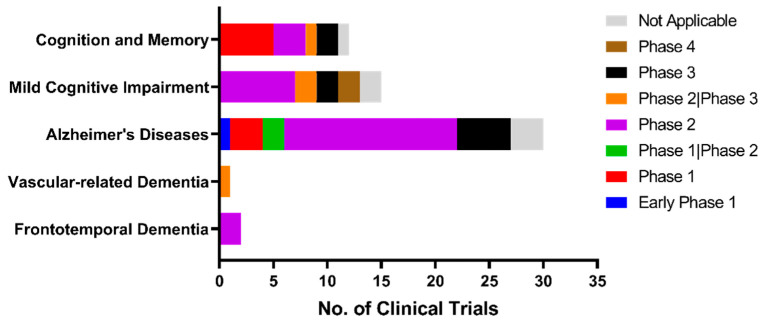
Targeted conditions and diseases in clinical trials repositioning anti-diabetic drugs for dementia treatment.

**Figure 4 ijms-24-11450-f004:**
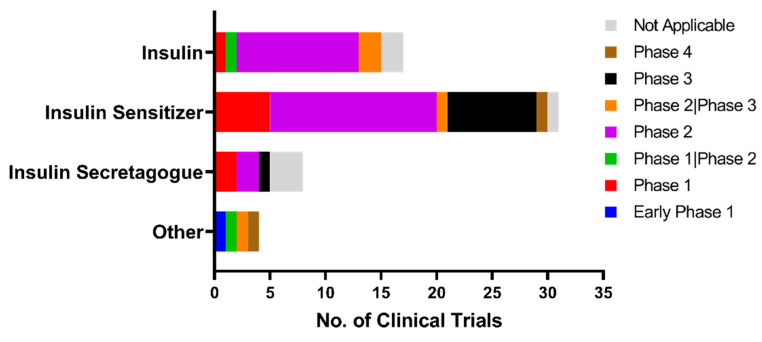
Classification of drug modes of action in clinical trials repositioning anti-diabetic drugs for dementia treatment.

## Data Availability

Not applicable.

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
