# Peer review of "Repositioning of Anti-Diabetic Drugs against Dementia: Insight from Molecular Perspectives to Clinical Trials"

_ijms, 2023, doi:10.3390/ijms241411450_

Round 1

Reviewer 1 Report

The manuscript "Repositioning of Anti-Diabetic Drugs Against Dementia: Insight from Molecular Perspectives to Clinical Trials" presents the relationship between biochemical substrates of DM2 and dementia and the potential efficacy of anti-diabetic drugs in this neurodegenerative condition. The authors detail the ongoing clinical trials of hypoglycemic drugs tested for dementia and discuss the respective trial phases and outcomes.

The paper is well written, and the detailed data is easy to follow due to the manuscript’s sound structure. 

However, I recommend that the authors add/mention in the first paragraph of section 4.4. Other Antidiabetic Drugs the SGLT-2 inhibitors involved in the research papers cited ast [121] and [123]. 

A minor observation: use consistently anti-diabetic (or antidiabetic) throughout the manuscritpt. 

Author Response

The manuscript "Repositioning of Anti-Diabetic Drugs Against Dementia: Insight from Molecular Perspectives to Clinical Trials" presents the relationship between biochemical substrates of DM2 and dementia and the potential efficacy of anti-diabetic drugs in this neurodegenerative condition. The authors detail the ongoing clinical trials of hypoglycemic drugs tested for dementia and discuss the respective trial phases and outcomes.

The paper is well written, and the detailed data is easy to follow due to the manuscript’s sound structure. 

However, I recommend that the authors add/mention in the first paragraph of section 4.4. Other Antidiabetic Drugs the SGLT-2 inhibitors involved in the research papers cited ast [121] and [123]. 

Response: We appreciate the valuable comments and totally agree the suggestion. We added sentences in the sub-section, which were marked as red-colored text.

A minor observation: use consistently anti-diabetic (or antidiabetic) throughout the manuscritpt. 

Response: We appreciate the comment and use "anti-diabetic" consistently throughout the manuscript.

Reviewer 2 Report

The manuscript entitled "Repositioning of Anti-Diabetic Drugs Against Dementia: Insight from Molecular Perspectives to Clinical Trials" is well written but minor language editing is required.

Mostly the targets are mentioned as applicable to Diabetes. A clear table of the drug should be given which are under preclinical phase or repurposed.

After these changes, manuscript may be accepted for publication.

The manuscript entitled "Repositioning of Anti-Diabetic Drugs Against Dementia: Insight from Molecular Perspectives to Clinical Trials" is well written but minor language editing is required.

Author Response

The manuscript entitled "Repositioning of Anti-Diabetic Drugs Against Dementia: Insight from Molecular Perspectives to Clinical Trials" is well written but minor language editing is required.

Mostly the targets are mentioned as applicable to Diabetes. A clear table of the drug should be given which are under preclinical phase or repurposed.

Response: We appreciate the favorable but critical comment. We added a table that summarize the mechanisms of action and pharmacological effects of anti-diabetic drugs exhibiting neuroprotection in preclinical study (Page 6).